# Unraveling the rheology of inverse vulcanized polymers

Derek J. Bischoff [ORCID][1], Taeheon Lee[2], Kyung-Seok Kang [ORCID][2], Jake Molineux[2], Wallace O'Neil Parker Jr.[3], Jeffrey Pyun [ORCID][2] ✉ & Michael E. Mackay [ORCID][1,4] ✉

Multiple relaxation times are used to capture the numerous stress relaxation modes found in bulk polymer melts. Herein, inverse vulcanization is used to synthesize high sulfur content (≥50 wt%) polymers that only need a single relaxation time to describe their stress relaxation. The S-S bonds in these organopolysulfides undergo dissociative bond exchange when exposed to elevated temperatures, making the bond exchange dominate the stress relaxation. Through the introduction of a dimeric norbornadiene crosslinker that improves thermomechanical properties, we show that it is possible for the Maxwell model of viscoelasticity to describe both dissociative covalent adaptable networks and living polymers, which is one of the few experimental realizations of a Maxwellian material. Rheological master curves utilizing time-temperature superposition were constructed using relaxation times as non-arbitrary horizontal shift factors. Despite advances in inverse vulcanization, this is the first complete characterization of the rheological properties of this class of unique polymeric material.

The development of stimuli-responsive macromolecules composed of reversible covalent bonds in the polymer backbone has prompted a paradigm shift in the understanding of rheological and thermo-mechanical properties of these materials. The vast majority of work in this field has investigated covalent adaptable networks (CANs), where bond reorganization within a crosslinked polymer network has a profound impact on the rheological properties by introducing additional stress relaxation modes[1–16]. To describe the frequency dependent, viscoelastic properties of polymer melts, multiple relaxation times are necessary to capture the complex interactions over a variety of length and time scales. Reptation theory espoused by de Gennes[17] predicts an infinite number of relaxation times even for a monodisperse polymer melt[18]. However, the stress relaxation of certain polymers with reversible bonds is dominated by the dynamic bond reorganization and may follow the simple Maxwell model of viscoelasticity to afford a single, dominant relaxation time. One of the very few examples of this behavior is the polymeric allotrope of elemental sulfur that is solely composed of dynamic S-S bonds[19].

The Maxwell model of viscoelasticity predicts a single relaxation time, $\tau_m$, that can be determined from the crossover frequency as $\tau_m = 1/\omega_m$, where $\omega_m$ is the frequency defined at the crossover of the storage modulus ($G'$) and loss modulus ($G''$). The model is often depicted as a spring and dashpot in series representing an ideal solid that follows Hooke's law of elasticity and an ideal liquid that follows Newton's law of viscosity, respectively (Fig. 1a). Conventional polymer melts have multiple relaxation times, or a spectrum, where no one mode is sufficiently dominant to fully describe its viscoelastic properties[20]. Generalized Maxwell models combine several Maxwell elements in parallel to better capture the physics in real materials, typically, one element is added per decade of frequency. The moduli of a single relaxation time Maxwell model are given by Eqs. 1 and 2

$$G' = G_N^0 (\omega\tau_m)^2 / \left(1 + (\omega\tau_m)^2\right) \qquad (1)$$

[1]Department of Materials Science and Engineering, University of Delaware, Newark, DE 19716, USA. [2]Department of Chemistry and Biochemistry & Wyant College of Optical Sciences, University of Arizona, Tucson, AZ 85721, USA. [3]Physical Chemistry Department, Eni S.p.A., San Donato Milanese 20097, Italy. [4]Department of Chemical and Biomolecular Engineering, University of Delaware, Newark, DE 19716, USA. ✉e-mail: jpyun@arizona.edu; mem@udel.edu

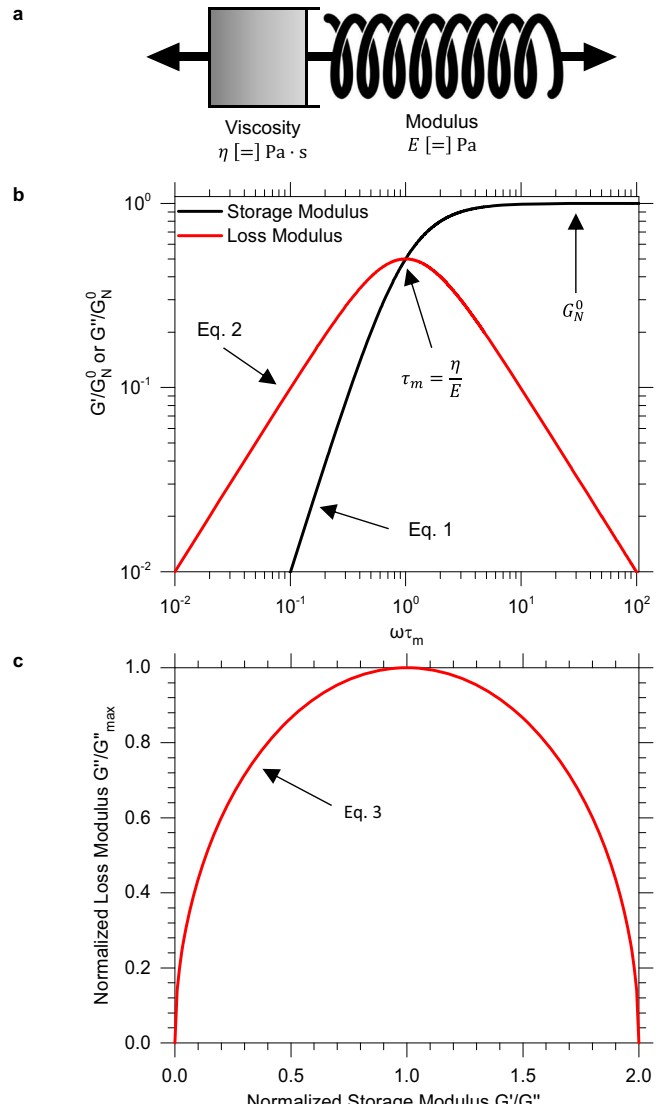

**Fig. 1 | Maxwellian material. a** a spring and dashpot combine in series to form a single Maxwell element to model a viscoelastic material's response to an applied stress (**b**) rheological master curve for a Maxwellian material highlighting the relaxation time as determined from the crossover in moduli (**c**) Cole-Cole representation for a Maxwellian material giving a semicircular shape.

$$G'' = G_N^0 (\omega \tau_m) / \left(1 + (\omega \tau_m)^2\right) \qquad (2)$$

where $G_N^0$ is the rubbery plateau modulus and $\omega$ is frequency (Fig. 1b). In the terminal regime (at longer timescales or lower frequencies), $G'$ and $G''$ are predicted to have a slope of two and one, respectively, on a log-log plot of moduli versus frequency. Notably, this master curve shows a unique maximum in $G''$ followed by a linear decay at higher frequencies, which is a distinctive feature of the Maxwell model, as most experimental polymer melts exhibit an inflection in $G''$ causing its value to rise at higher frequency. The Maxwell relationship can be written to eliminate frequency dependence as shown in Eq. 3. This equation is dimensionless with $G''_{max} = G_N^0 / 2$ since the maximum occurs at a frequency of $1/\tau_m$.

$$G''^* = \sqrt{G'^* \times (2 - G'^*)} \qquad (3)$$

This equation was produced by normalizing the moduli with $G''_{max}$ and accounts for the * notation. The normalization technique was chosen since the maximum of $G''$ is easily identifiable in the rheological data. Upon plotting Eq. 3, a Cole-Cole representation[21] is achieved whereby the normalized imaginary component, $G''$, is plotted against the normalized real part, $G'$, of the complex modulus on a linear scale (Fig. 1c). The Maxwell model predicts a semicircular shape, affording a convenient graphical method to determine if a material follows the Maxwell model and has a single, Debye-like[22] relaxation mode. Polymeric materials that exhibit a single relaxation time and yield a relaxation spectrum as shown in Fig. 1b, c are highly unusual in polymer science and are herein referred to as Maxwellian polymers.

Independent of the Maxwell model and the field of CANs, Cates[23] defined the term living polymers that describe linear polymer chains with reversible main chain scission on experimental time scales with presumably fast bond exchange to suppress depolymerization (Fig. 2a). These rheologically living polymers are definitionally distinct from polymers possessing active propagating chain ends formed via living anionic polymerizations or related controlled polymerizations[24,25]. Cates' model was developed to specifically address surfactant organization in microemulsions and micelles as well as homogenous equilibrium polymers such as liquid sulfur that has polymerized above its $\lambda$-transition[23,26]. Cates' theory has been successfully used to experimentally describe the viscoelastic behavior of worm-like micelles[21,27–34] that break and diffuse during deformation, however, the realization of bulk living polymer melts remains elusive, with the exception of pure sulfur[22]. Application of Cates' model yields the curious prediction that the geometric mean of the breakage and reptation times, under certain conditions, dominate the rheology of living polymers and thus exhibit a single relaxation time. As such, the experimental rheological properties would follow the Maxwell model when the bond scission/recombination kinetics are fast compared to the viscoelastic relaxation time. Notably certain polysulfide rubbers show Maxwellian stress relaxation[35,36]. It should be noted that even a living polymer system has other relaxation times associated with submolecular modes that can be absorbed into Cates' model[37,38], however, the dominant mode is predicted to be associated with the living relaxation time. Hence, living polymers can be described as Maxwellian polymers that possess dynamic reversible bonds within a linear, or non-crosslinked polymer architecture. Therefore, living polymers represent a new class of condensed matter with dynamic bonds that can be simply thought of as non-crosslinked analogues of CAN materials (Fig. 2b).

With both the Maxwell and living polymer models in mind, we set forth to engineer a bulk, robust polymeric material that could be described by a single relaxation time, in this case with inexpensive, dynamic covalent S-S bonds made from elemental sulfur ($S_8$). The inverse vulcanization process, developed by Pyun et al.[39] is ideally suited for this purpose where $S_8$ is used as the monomeric reaction medium for bulk homolytic ring-opening polymerizations with organic unsaturated monomers to afford inverse vulcanized polymers or organopolysulfides[40–44]. Inverse vulcanized polysulfides inherently contain dynamic covalent S-S bonds in the copolymer backbones offering a route to dissociative bond exchange without the need to explicitly engineer reversibly covalent moieties. However, the intractable nature of these organopolysulfides stemming from their poor solubility in common solvents and dynamic S-S bond exchange has complicated detailed experimental and mechanistic characterization methods to understand their true microstructure. These inherent challenges have prevented unambiguous classification of the inverse vulcanized polysulfides as thermoplastics or thermosets (or neither). To date, the only rheological study of inverse vulcanized polysulfides was conducted by Mackay and Pyun et al.[45] in which polymers containing a higher sulfur rank were found to be more dynamic when exposed to heat and shear deformation. Ultimately, no frequency sweeps or master curves were included leaving an experimental need to probe the terminal region and definitive flow behavior.

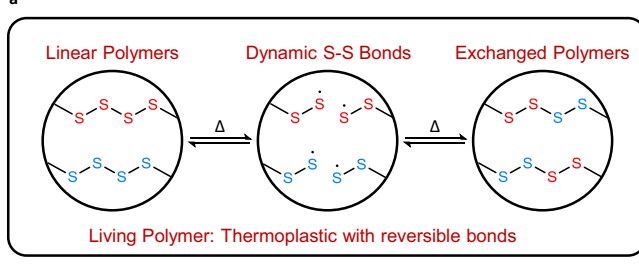

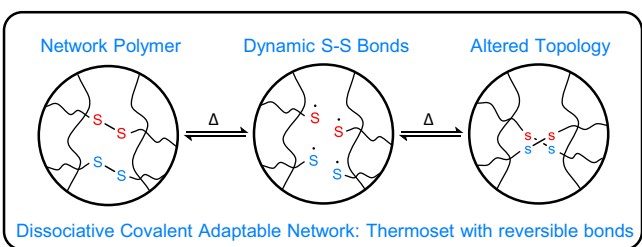

**Fig. 2 | Comparison of polymer types. a** living polymers are linear polymers with exchangeable bonds that undergo reversible chain scission reactions and recombine with other dynamic linear chains (**b**) dissociative covalent adaptable networks (CANs) are covalently crosslinked polymer networks with exchangeable bonds that when exposed to a stimulus (i.e., heat) alter their topology and enable stress relaxation/flow.

Herein, we report the first detailed rheological analysis of inverse vulcanized polysulfides of varying sulfur composition, organic comonomer ratio, and polymer architecture. Using small amplitude oscillatory shear (SAOS) rheology, we demonstrate that through control of co- and terpolymer composition, inverse vulcanized polysulfides can be engineered to behave as classical thermoplastics or dissociative CANs by altering crosslinking density. This study also indicates that inverse vulcanized polysulfides of certain compositions are also Maxwellian and living polymers, which is one of first experimental realizations of these rheological models. The key molecular causality of these bulk rheological properties arises from the significant number of dynamic S-S bonds present in these materials which is an important finding in the fields of polymer chemistry and materials science.

## Results and Discussion

A series of co- and terpolymers for these rheological studies were prepared via inverse vulcanization of $S_8$ (50 wt% or 70 wt%) with combinations of styrene (Sty), 1,3-diisopropenylbenzene (DIB), or dimeric norbornadiene (NBD2) to determine design criteria for Maxwellian behavior. The terpolymer poly($S_{50}$-$r$-DIB$_{25}$-$r$-NBD2$_{25}$) contains 50 wt% sulfur, 25 wt% DIB, and 25 wt% NBD2, for example. Poly($S$-$r$-DIB) copolymers prepared by inverse vulcanization were previously thought to be branched or network polymers as determined by curing conditions and therefore were assumed to be dissociative CANs[46]. However, a recent mechanistic report by Pyun et al.[47]. has proven that poly($S$-$r$-DIB) is primarily composed of linear bis-cumyl units, which disqualifies these copolymers as CANs. Alternatively, poly($S$-$r$-NBD2) was more recently prepared where NBD2 was presumed to serve as a crosslinking agent to improve thermomechanical properties[48]. The glass transition temperature ($T_g$) of poly($S$-$r$-NBD2) is -100 °C, which is vastly improved over that of -30 °C for poly($S$-$r$-DIB), preventing reflow issues when molded or processed in the solid state[48]. The reaction scheme for all synthesized materials is shown in Fig. 3 along with revised structures for DIB units incorporated into poly($S$-$r$-DIB-$r$-NBD2), where the idealized microstructure of DIB is replaced with a bis-cumyl type unit[47]. Solid-state $^{13}$C nuclear magnetic resonance (NMR) spectroscopy was conducted on the terpolymers to confirm the

incorporation of both cyclic aliphatic and aromatic moieties (Supplementary Figs. 3–4). NMR spectroscopic, mechanistic, and computational evidence for this inverse vulcanization mechanism has been discussed elsewhere[47], but indicates that copolymerization of $S_8$ with either Sty or DIB affords linear type units, while NBD2 increases crosslinking density.

Figure 4 shows the first bulk polymeric organopolysulfide material that can be classified as both a living and a Maxwellian polymer: poly($S_{70}$-$r$-DIB$_{30}$). Frequency sweep data collected using SAOS rheology provides the frequency dependent $G'$ and $G''$ values at various temperatures. Replotting the data in a Cole-Cole representation, shown in Fig. 4a, exhibits the characteristic semicircular shape as discussed in Fig. 1c for a Maxwellian material. Following the same procedure for each temperature that experimentally shows a maximum in $G''$ enables the clear realization that the material has a single dominant relaxation mode. At higher values of normalized storage modulus, small deviations from the semicircle are attributed to Rouse-like motions and other (sub)molecular processes that contribute to relaxation at shorter time scales[23,38]. This behavior is also observed in elemental sulfur as described by Stashick and Marriott[22]. By dividing Eq. 1 by Eq. 2, the linear relationship in Eq. 4 is obtained that can be plotted to obtain $\tau_m$ as the slope.

$$G'/G'' = \tau_m \times \omega \qquad (4)$$

Figure 4b shows the linear fit for each temperature at which a maximum in $G''$ is observed. To keep in the spirit of the Maxwell model, a linear line of best fit with its intercept forced through zero is used. Using a forced zero intercept (0.149 s) showed little change in the relaxation time compared to not forcing the intercept (0.145 s) for the data at 160 °C. (Supplementary Figs. 30 and 31 for poly($S_{50}$-$r$-DIB$_{50}$)). To capture $G'$ and $G''$ data over a wider frequency range than is experimentally possible, it is customary to employ time-temperature superposition (TTS). The Maxwell relaxation times were used as horizontal shift factors to generate the master curve (Fig. 4c) and were not arbitrarily determined. The vertical shift factors are then determined by empirically shifting the data to obtain good overlay amongst the different temperatures used to construct the dimensionless master curve. TTS often fails to describe the rheology of dissociative CANs made entirely from organic moieties. The resulting master curve shows good agreement after shifting except after passing the maximum in G″ with dimensionless frequency. It was hypothesized that this could be caused by solubilized sulfur in the polymer leading to bloom, however, deviations are still observed in poly($S_{50}$-$r$-DIB$_{50}$) even when new material is used for each oscillatory frequency sweep that drastically reduced the exposure time of the material to elevated temperatures (Supplementary Fig. 32). The empirical vertical shift factors are significant in the material, which span from 0.36 to 3.2 over only a 30 °C temperature range as shown in Fig. 4d. To our knowledge, this is a rare example of a soft material requiring these large shift factors, the validity of which is demonstrated by the good fit of the dimensionless master curve.

The dominant stress relaxation mechanism is through rapid S-S bond exchange compared to reptation in agreement with Cates[26] as well as Stashick and Marriott[22]. The single relaxation time predicted by the adherence to the Maxwell model indicates that the chemorheological relaxation mechanism (bond scission/recombination) is independent of the polymer chain length and therefore of the diffusional reptation time. As the rank of sulfur (i.e., the number of consecutive S-S bonds in a chain) increases, the S-S bonds become more dynamic in nature[49], where the S-S bond farthest from the C-S bonds connecting to the organic comonomer possesses the lowest bond dissociation energy. The S-S exchange reaction is dissociative, meaning that a long linear polymer chain can transiently break into two shorter, distinct polymer chains without needing to interact with another chemical

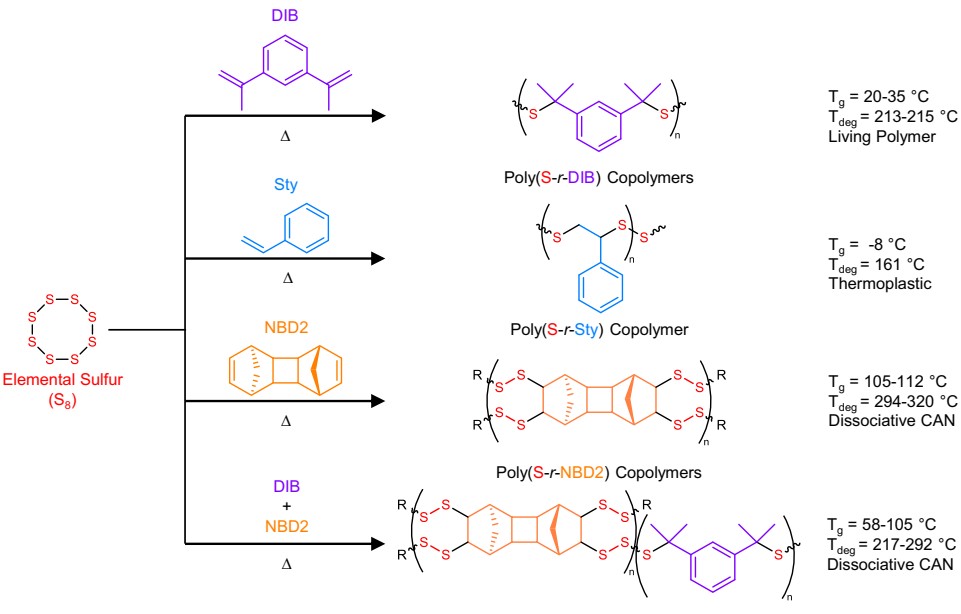

**Fig. 3 | Synthesis of inverse vulcanized polysulfides.** Reaction scheme to synthesize the inverse vulcanized copolymers and terpolymers using $S_8$, Sty, DIB, and NBD2. Through the control of composition, the polysulfides can behave as thermoplastics, Maxwellian/living polymers, or dissociative covalent adaptable networks.

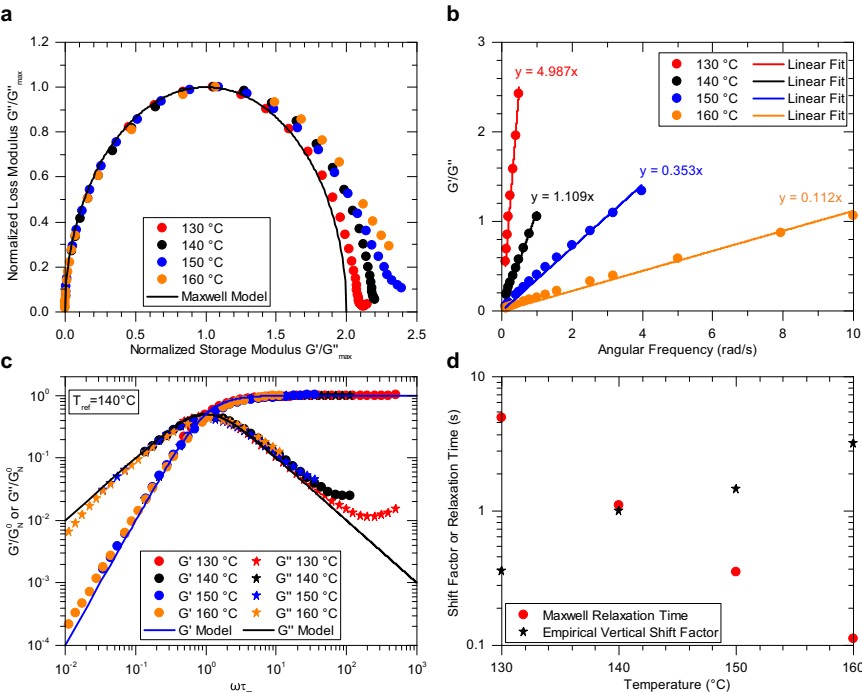

**Fig. 4 | Adherence of the poly($S_{70}$-r-DIB$_{30}$) copolymer to the Maxwell model of viscoelasticity. a** Cole-Cole plot showing the semicircular shape indicating that the material is Maxwellian and dominated by a single relaxation mode. **b** Determination of the Maxwell relaxation time at each temperature displaying a maximum in loss modulus as predicted by the Maxwell model. **c** Dimensionless master curve utilizing Maxwell relaxation times for horizontal shifting and empirical shifting along the vertical axis. Solid black and blue lines show the Maxwell model. **d** Resulting shift factors showing the importance of the vertical shift factor.

moiety belonging to another chain (Fig. 2a). As such, when the dynamic covalent chemistry is activated by thermal stimulus, the transient average molecular weight of the polymer decreases before subsequent thiyl radical coupling upon cooling. As temperature increases, the equilibrium constant begins to shift more and more towards the dissociated, oligomeric form which is not captured by Cates' theory, and it does not account for the rapid decrease in plateau modulus with temperature, discussed next. The combination of the single relaxation time, the S-S bond exchange, and the linear polymer architecture indicate that poly($S_{70}$-r-DIB$_{30}$) is indeed a living polymer. Likewise, poly($S_{50}$-r-DIB$_{50}$) exhibits similar rheological properties as the poly($S_{70}$-r-DIB$_{30}$) composition and is therefore both Maxwellian and living despite the difference in sulfur rank (Supplementary Fig. 30).

Figure 5 was constructed to show the original, unshifted $G'$ data for poly($S_{70}$-$r$-$DIB_{30}$) as a representative example of these organopolysulfides which prominently illustrates the variance of rubbery plateau modulus as a function of temperature providing insight into the large vertical shift factors observed experimentally (Supplementary Fig. 43 for poly($S_{50}$-$r$-$DIB_{50}$) which exhibits identical behavior). Not seen in this plot is the increase in moduli associated with shorter length scales and glassy dynamics at higher frequency that superimpose in the overall master curve (Supplementary Fig. 18). Since Cates' model did not account for the reversible equilibrium between S-S bonds and thiyl radicals, the strong temperature dependence of the rubbery plateau modulus was not captured, explaining why the vertical shift factor is so large relative to traditional polymer melts where the shift factor is usually within the range of 0.9 to 1.1 (Supplementary Fig. 27 for traditional polystyrene). The large vertical shift factors, essentially the ratio of the rubbery plateau modulus at a given temperature compared to that at the reference temperature, are shown in Fig. 4d. Figure 5 also shows the terminal behavior at different temperatures. At higher temperatures, terminal behavior is achieved on a shorter timescale which is a feature of CANs as the kinetics of the bond exchange reaction increases with temperature. The trend of terminal behavior occurring at shorter timescales also occurs in traditional thermoplastic melts due to the increased reptation rate of polymer chains, but the effect is enhanced by the increasing rate of bond exchange reactions.

The importance of the thermal stimulus in activating the S-S bond exchange is demonstrated using the soluble, linear thermoplastic poly($S_{50}$-$r$-$Sty_{50}$), which possesses a subambient $T_g$ of $-8$ °C (Fig. 3)[50]. The material was studied between $-20$ °C and $10$ °C utilizing liquid nitrogen to realize the subambient temperatures. The master curve constructed using standard TTS where horizontal shift factors are arbitrarily determined, at a reference temperature of $-10$ °C, in Fig. 6a shows a crossover between $G'$ and $G''$ giving a relaxation time of $\tau = [0.872\,\text{rad/s}]^{-1} = 1.15$ sec, a terminal viscosity of ~$1.54 \times 10^8$ Pa·s, and a rubbery plateau modulus of 171 MPa. The rheological behavior of poly($S_{50}$-$r$-$Sty_{50}$) is not Maxwellian due to the low $T_g$ of this sulfur copolymer, enabling terminal flow before significant S-S bond exchange occurs. The S-S bonds have not been observed to be dynamic below ~90 °C in poly(S-$r$-DIB)[46], but have been reported at

110 °C[45,46,51,52]. S-S bond scission in poly(S-$r$-Sty) is exploited in dynamic covalent polymerizations at temperatures between 110-130 °C[52]. Moreover, the mechanism for stress relaxation is likely through diffusion/reptation and not S-S bond exchange. The horizontal and vertical shift factors (Fig. 6b) were determined empirically and the horizontal shift factors follow the standard Williams–Landel–Ferry (WLF) model[53]. The vertical shift factors do not vary significantly over a 30 °C temperature change, in contrast to the living polymers discussed previously since the dissociative bond exchange is not active, making the average molecular weight of the polymer a constant. The non-Maxwellian behavior is further emphasized in Fig. 6c where the data shown using a Cole-Cole representation does not form a semicircular shape. Therefore, the lack of thermally activated S-S bond exchange combined with the fact that poly(S-$r$-Sty) is linear makes the material a traditional thermoplastic under ambient conditions. To further highlight the tunable rheological properties between poly(S-$r$-Sty) and poly(S-$r$-DIB), a series of linear poly(S-$r$-DIB-$r$-Sty) terpolymers was synthesized via inverse vulcanization. The inclusion of more rigid DIB units into the poly(S-$r$-Sty) backbone predictably raised the $T_g$ of the terpolymers in turn suppressing terminal flow before the onset of S-S bond exchange. Hence, we demonstrate the recovery of Maxwellian behavior at the appropriate terpolymer composition of poly($S_{50}$-$r$-$DIB_{40}$-$r$-$Sty_{10}$) (Supplementary Fig. 36) and $T_g$ (17 °C; Supplementary Fig. 11). Furthermore, the synthesis of terpolymers with higher Sty compositions (poly($S_{50}$-$r$-$DIB_{25}$-$r$-$Sty_{25}$) and poly($S_{50}$-$r$-$DIB_{10}$-$r$-$Sty_{40}$)) lowered the $T_g$ of these materials ($T_g = 0$ °C and $-18$ °C, respectively) so that terminal flow was observed before S-S bond exchange was activated and Maxwellian relaxation was no longer observed (Supplementary Figs. 22–25) which is also true in poly($S_{50}$-$r$-$Sty_{50}$).

Having established that Maxwellian behavior is obtainable in linear living polymers of poly(S-$r$-DIB) due to the thermal activation of the S-S bond exchange, rheological characterization of the crosslinked organopolysulfide poly(S-$r$-NBD2) was conducted. In contrast to styrene or DIB that possess a monomer functionality of two or less when used for inverse vulcanization, we presume that the two olefinic units of NBD2 can form four C-S bonds thereby affording NBD2 an effective functionality of four, which is qualitatively supported by solid-state NMR spectroscopy of poly(S-$r$-NBD2) (Supplementary Figs. 3 and 4). Compared to poly($S_{70}$-$r$-$DIB_{30}$) or poly($S_{50}$-$r$-$DIB_{50}$), poly($S_{50}$-$r$-$NBD2_{50}$) has a lower sulfur rank due to the higher monomer functionality resulting in stronger S-S bonds that are less likely to dissociate. Higher bond dissociation energies are observed for shorter S-S chains, greatly reducing the number of S-S bond exchange events in the crosslinked organopolysulfide[54]. The master curve shown in Fig. 7a, constructed using standard TTS, does not show Maxwellian behavior as it does not share the master curve shape shown in Fig. 1b. The Cole-Cole plot of the master curve (Fig. 7c) shows an incomplete semicircle and therefore is not a Maxwellian polymer, although very large vertical shift factors are required to shift the poly($S_{50}$-$r$-$NBD2_{50}$) data to form the master curve (Fig. 7b).

Interestingly, the master curve for the crosslinked poly($S_{50}$-$r$-$NBD2_{50}$) does have a terminal region, indicating that complete stress relaxation is possible over a sufficient time scale. The terminal relaxation behavior observed here indicates that this copolymer is a dissociative covalent adaptable network whereby the terminal relaxation is achieved via reversible S-S bond scission. A typical crosslinked network where the crosslinking density is constant regardless of temperature will show an equilibrium modulus at low frequency that grows in value with the degree of crosslinking[20]. Only a single crossover is observed giving $\tau = [1.339\,\text{rad/s}]^{-1} = 0.747$ sec ($T_{ref} = 230$ °C), along with a terminal viscosity of ~$1.14 \times 10^8$ Pa·s. The rubbery plateau modulus for poly($S_{50}$-$r$-$NBD2_{50}$) is 99.3 MPa compared to only 4.57 MPa for poly($S_{70}$-$r$-$DIB_{30}$). The results for poly($S_{70}$-$r$-$NBD2_{30}$) are shown in Supplementary Fig. 35 which reveal Maxwellian behavior due to the higher sulfur content that imparts more

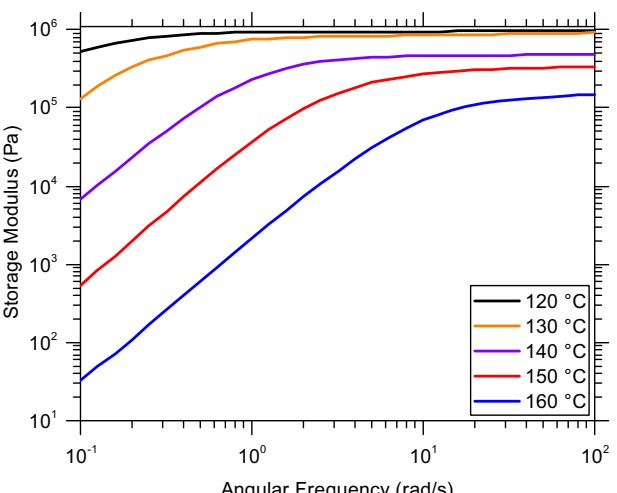

**Fig. 5 | Terminal region of the Maxwellian copolymer poly($S_{70}$-$r$-$DIB_{30}$).** The drop off in storage modulus, here due to the organopolysulfide exchange reaction, is a feature of polymers containing dynamic covalent bonds in general that show a more rapid stress relaxation as the rate of reaction increases with temperature, thus shortening the rubbery plateau region. This unshifted data also emphasizes the importance of the vertical shift factor as its value increases as the equilibrium constant shifts towards the dissociated thiyl radicals.

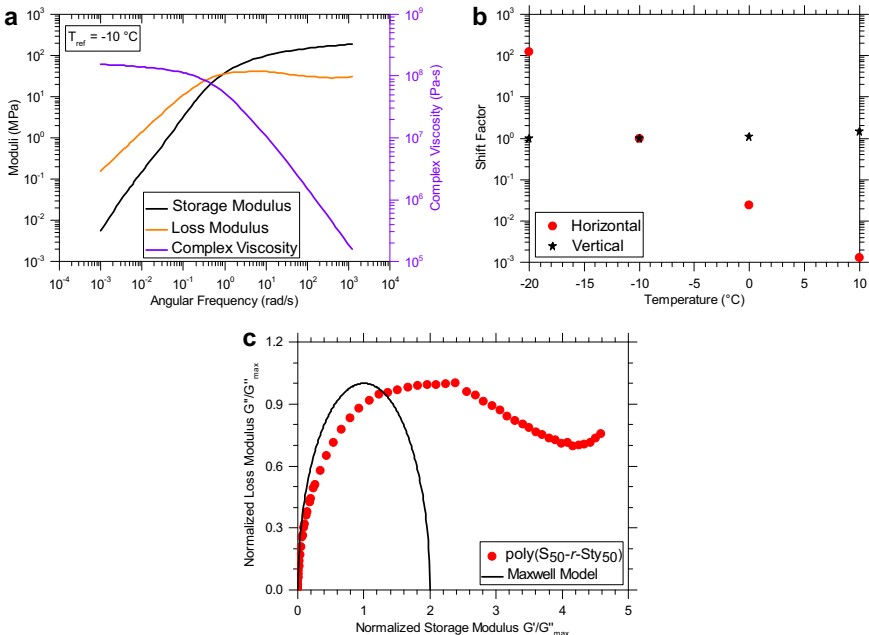

**Fig. 6 | Copolymer poly(S₅₀-r-Sty₅₀) rheological results. a** master curve for linear poly(S$_{50}$-r-Sty$_{50}$) at T$_{ref}$ = −10 °C (**b**) the horizontal and vertical shift factors were determined empirically using standard rheological techniques. **c** Cole-Cole representation showing that the material is not a Maxwellian polymer.

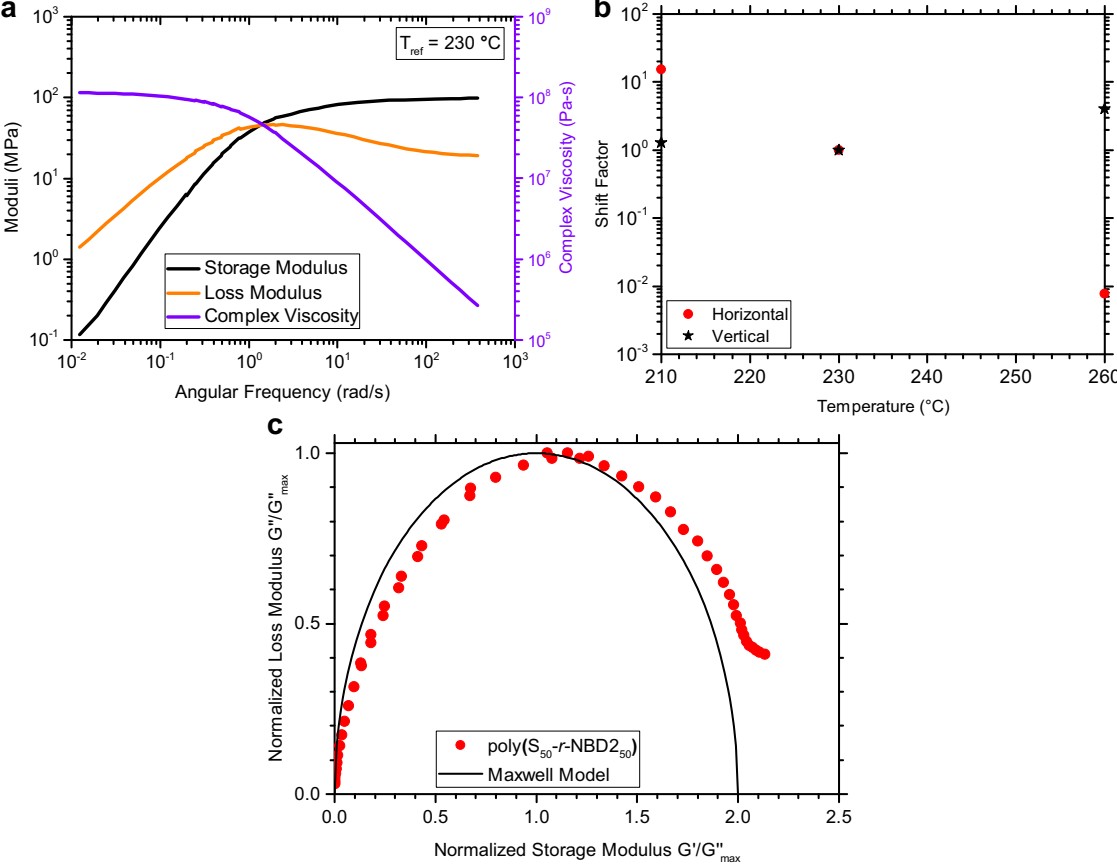

**Fig. 7 | Crosslinked copolymer poly(S₅₀-r-NBD2₅₀) rheological results. a** master curve of at T$_{ref}$ = 230 °C (**b**) shift factors used to make the master curve following standard TTS (**c**) Cole-Cole plot of the shifted data revealing an incomplete semicircular shape and therefore non-Maxwellian behavior.

dynamic S-S bonds that outweigh the restriction from the NBD2 crosslinks.

Thus far, we have shown that the linear copolymers poly($S_{70}$-$r$-DIB$_{30}$) and poly($S_{50}$-$r$-DIB$_{50}$) are living, Maxwellian polymers and that the crosslinked network poly($S_{50}$-$r$-NBD2$_{50}$) is not Maxwellian, but rheologically behaves as a dissociative CAN. Terpolymers of poly($S$-$r$-DIB-$r$-NBD2) with varying sulfur (50 wt%, 70 wt%), DIB (10–40 wt%), and NBD2 (10–40 wt%) composition were prepared to ascertain the rheological effects of NBD2 comonomer units and prepare a thermo-mechanically enhanced, melt-processable material. While the poly($S_{70}$-$r$-DIB$_{30}$) copolymer possesses favorable optical properties, the low $T_g$ of this material precludes its use for plastic optics[55]. Fig. 8 shows the rheological results of the terpolymer poly($S_{70}$-$r$-DIB$_{15}$-$r$-NBD2$_{15}$). Unlike the highly crosslinked poly($S_{50}$-$r$-NBD2$_{50}$) just discussed, the terpoly-mer shows the recovery of Maxwellian stress relaxation behavior. The adherence to the Maxwell model was confirmed by the semicircular fit of Cole-Cole plot in Fig. 8a. Determination of the Maxwell relaxation time at each temperature (Fig. 8b) enabled the construction of the master curve in Fig. 8c which afforded good shifting of the data obtained between 150-180 °C using a reference temperature of 160 °C. Again, the Maxwell relaxation times are used to shift horizontally. Large vertical shift factors were also required for this terpolymer as observed for DIB- and NBD2-containing inverse vulcanized copoly-mers (Fig. 8d). The introduction of the DIB units into the crosslinked polymer network increases the length of the linear segments between crosslinks, along with the recovery of Maxwellian rheological behavior. Due to the presence of NBD2 crosslinker in the terpolymer, the material is crosslinked and is therefore a Maxwellian CAN.

The other terpolymer CAN compositions (both poly($S_{50}$-$r$-DIB$_{40}$-$r$-NBD2$_{10}$) and poly($S_{50}$-$r$-DIB$_{25}$-$r$-NBD2$_{25}$)) that show Maxwellian beha-vior are shown in Supplementary Figs. 33 and 34. These compositions behave similarly in that they are Maxwellian, shift to form a dimen-sionless master curve using relaxation times as horizontal shift factors, and they have large vertical shift factors. The most highly crosslinked terpolymer composition tested, poly($S_{50}$-$r$-DIB$_{10}$-$r$-NBD2$_{40}$), does not reveal Maxwellian behavior (Supplementary Fig. 16) and cannot be

shifted using the same procedure outlined above and behaved as a dissociative CAN material as discussed for poly($S_{50}$-$r$-NBD2$_{50}$). The master curves of all materials tested, generated using standard, empirical TTS, are shown in Supplementary Figs. 13–22 and 24. The presence of a terminal region in the master curves of all organopoly-sulfides tested indicates the materials undergo complete relaxation and imply that polymer processing techniques like extrusion are amenable. There is expected to be a maximum amount of NBD2 that can be added before a departure from Maxwellian behavior is observed, likely due to the shortening of linear S-S chains and asso-ciated higher bond dissociation energies. The limit appears to be between 25 and 40 wt% NBD2 since poly($S_{50}$-$r$-DIB$_{25}$-$r$-NBD2$_{25}$) is Maxwellian, but poly($S_{50}$-$r$-DIB$_{10}$-$r$-NBD2$_{40}$) is not. We have alluded to the point that the thermomechanical properties of the terpolymers are tunable, and those trends will now be discussed.

Aside from the tuning of the Maxwellian behavior, the thermo-mechanical properties of these terpolymers were likewise controllable by adjusting composition. The reflow issues stemming from the rela-tively poor thermomechanical properties of living poly($S$-$r$-DIB) were improved by using NBD2, however, the CAN copolymers poly($S$-$r$-NBD2) are phenomenologically thermosets[48]. Because of this, the addition of NBD2 comonomer to poly($S$-$r$-DIB) was employed to raise the $T_g$ and degradation temperature ($T_{deg}$) of these materials while retaining melt processability. Thermogravimetric analysis (TGA) for sulfur co- and terpolymers was conducted to determine the $T_{deg}$ of each material, as determined by a 5% drop in mass. These results also provide the upper limit for the differential scanning calorimetry (DSC) heating ramps and rheological testing (Supplementary Figs. 5–7). The soluble oligomeric poly($S_{50}$-$r$-Sty$_{50}$) exhibited the lowest $T_{deg}$ (161 °C) and the least amount of char yield, only 0.03%. For polymers com-prised of 50 wt% and 70 wt% sulfur, $T_{deg}$ values were observed to increase with higher NBD2 comonomer content due to the higher crosslinking density and reduced sulfur rank in the polymer network. All inverse vulcanized organopolysulfides containing DIB and/or NBD2 show a higher $T_{deg}$ than elemental sulfur (194 °C, Supplementary Fig. 8), along with an increase in the char yield, which is also indicative

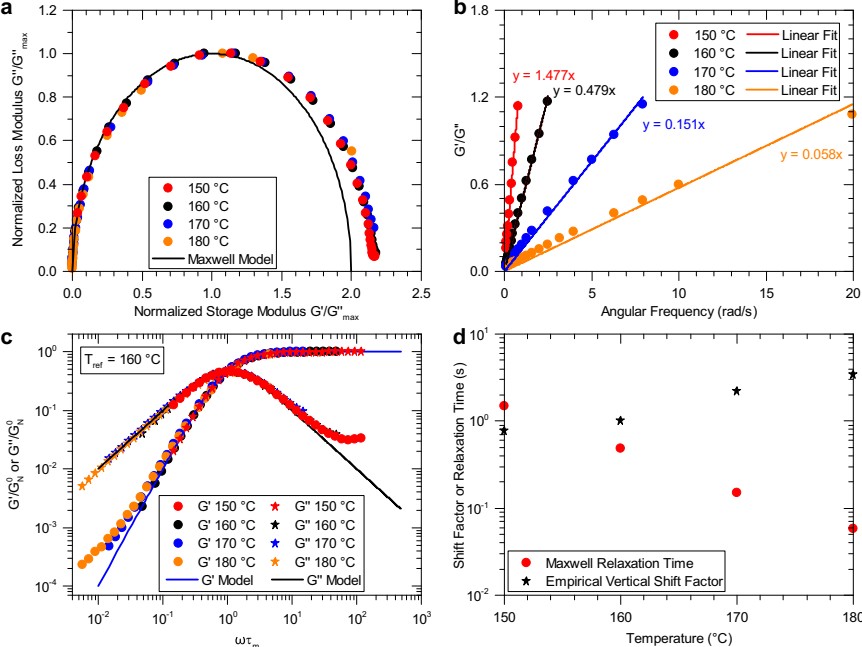

**Fig. 8 | Adherence of the terpolymer poly($S_{70}$-$r$-DIB$_{15}$-$r$-NBD2$_{15}$) to the Maxwell model of viscoelasticity. a** Cole-Cole plot showing the semicircular shape indi-cating that the material is Maxwellian. **b** Determination of Maxwell relaxation time at each temperature displaying a maximum in loss modulus as predicted by the Maxwell model. **c** Dimensionless master curve utilizing Maxwell relaxation times for horizontal shifting and empirical shifting along the vertical axis; solid black and blue lines show the Maxwell model. **d** Resulting shift factors showing the impor-tance of the vertical shift factor.

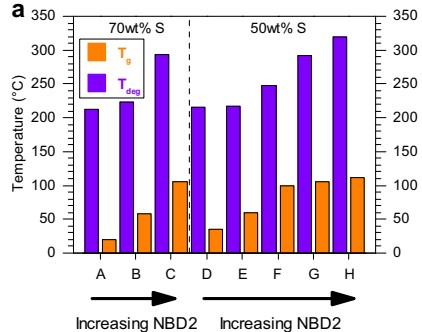
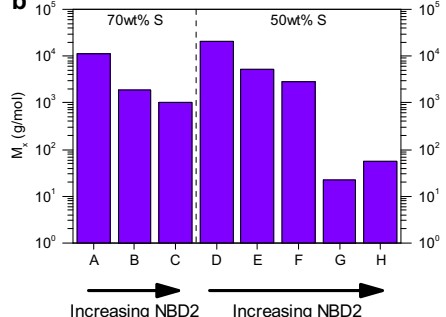

**Fig. 9 | Trends in thermal properties and molecular weight between topological constraints. a** Degradation and glass transition temperatures for each polymer. A 5% mass loss and second heating scans were used to determine the degradation and glass transition temperatures, respectively. **b** Values of $G_N^0$ used in the calculations are obtained from the master curves (Supplementary

Figs. 13–21) at their respective reference temperatures when $G''$ goes through a minimum. A = poly($S_{70}$-$r$-DIB$_{30}$); B = poly($S_{70}$-$r$-DIB$_{15}$-$r$-NBD2$_{15}$); C= poly($S_{70}$-$r$-NBD2$_{30}$); D = poly($S_{50}$-$r$-DIB$_{50}$); E = poly($S_{50}$-$r$-DIB$_{40}$-$r$-NBD2$_{10}$); F = poly($S_{50}$-$r$-DIB$_{25}$-$r$-NBD2$_{25}$); G = poly($S_{50}$-$r$-DIB$_{10}$-$r$-NBD2$_{40}$); H = poly($S_{50}$-$r$-NBD2$_{50}$).

of a crosslinked system[56]. Poly($S_{50}$-$r$-DIB$_{10}$-$r$-NBD2$_{40}$) and poly($S_{50}$-$r$-NBD2$_{50}$) had similar char yields from TGA measurements most likely due to the similar NBD2 content in these materials. These structure-thermal property trends were further corroborated with the calculation of molecular weight between topological constraints discussed later for these two systems.

DSC was utilized to determine the $T_g$ of sulfur co- and terpolymers, where materials possessing 50 wt% sulfur (Supplementary Fig. 9) show a trend of increasing $T_g$ with higher NBD2 comonomer compositions. A similar trend was observed with the 70 wt% series where increasing $T_g$ correlates with higher NBD2 content (Supplementary Fig. 10). This general trend is consistent with other cross-linkers used in polysulfides[39,57,58] and more broadly with increasing crosslinking density in traditional thermosets[59]. Interestingly, the $T_g$ of materials with a fixed 50 wt% sulfur exhibited a broad, tunable range from 35 to 112 °C by varying the ratio of NBD2 crosslinker with DIB comonomers. Likewise, $T_g$ was tunable between 20 and 105 °C with the 70 wt% sulfur materials while varying feed ratios of DIB and NBD2 (Fig. 9a). Additionally, the combination of DIB and NBD2 comonomers can successfully consume $S_8$ to (nearly) complete conversion over the compositional range tested which effectively eliminates residual $S_8$ blooming in the final material. Crystalline $S_8$ following bloom would be observable by a melting peak in DSC near ~113 °C corresponding to the melting of sulfur (Supplementary Fig. 12). The linear polymer poly($S_{50}$-$r$-Sty$_{50}$) exhibited a low $T_g$ value (-8 °C) due to the lack of crosslinks, increasing the overall mobility of the material (Supplementary Fig. 11). The low $T_g$ value makes it a viscous liquid at room temperature and thus the thermally activated S-S bond exchange is not needed for stress relaxation to occur in the material. All of the polysulfide polymers display a higher $T_g$ than that of polymeric sulfur ($T_g = -30$ °C), referring to the quick-quenched morphology from 200 °C[60].

Since adherence to the Maxwell model is a function of cross-linking density, it is useful to understand how the crosslinking density changes with NBD2 composition. Using the plateau modulus from the master curves (marked by a star in Supplementary Figs. 13–22 and 24) of sulfur co- and terpolymers, the molecular weight between topological constraints ($M_x$) (i.e., crosslinks/entanglements) was calculated for each material. These effective $M_x$ can be determined using Eq. 5 where $\rho$ is the polymer density, $R$ is the ideal gas constant, and $T$ is temperature[20]. The value of $G'$ in the rubbery plateau where $G''$ is at a minimum is used as the plateau modulus, $G_N^0$.

$$G_N^0 = \frac{\rho RT}{M_x} \qquad (5)$$

The resulting values of $M_x$ are displayed in Fig. 9b assuming that the density of each polymer is the same at 1.3 g/cc, as determined by compression molding a cylinder of poly($S_{50}$-$r$-DIB$_{50}$) material and weighing it. Within the 50 wt% sulfur compositions, the value of $M_x$ decreases accordingly with an increasing amount of NBD2 crosslinker with the exception of poly($S_{50}$-$r$-DIB$_{10}$-$r$-NBD2$_{40}$). Similarly, within the 70 wt% sulfur compositions, $M_x$ decreases with an increasing amount of NBD2. An increase in crosslinking density is synonymous with decreasing $M_x$ between topological constraints. Equation 5 gives very low values for $M_x$ in the case of poly($S_{50}$-$r$-DIB$_{10}$-$r$-NBD2$_{40}$) and poly($S_{50}$-$r$-NBD2$_{50}$) that are on the order of a few atoms between crosslinks. Physically these results are nonsensical and are most likely due to excluding the effects of dynamic bonds in rubber network theory. The incorporation of such effects and insights from a microscopic perspective[61] in the development of a new dynamic framework may be a worthwhile theoretical endeavor to undertake. It is also likely that the network is saturated with crosslinks in these two highly crosslinked systems which makes it difficult to differentiate the two.

The stress relaxation in the polysulfides is due to S-S bond rearrangement, however, due to differences in sulfur rank and chemical structure, the temperature dependence of the horizontal and vertical shift factors may be different and require investigation. The horizontal shift factors ($\tau_m$), determined from fitting the Maxwell model to the data, rely on the fact that one relaxation mode is dominant and is attributed to the S-S bond exchange reactions and therefore have a single temperature dependency. As shown above, this time scale was determined by fitting the low frequency data and is not an arbitrary constant. Its temperature dependence was found from the Arrhenius equation:

$$\tau_m \text{ or } b_T = A \times \exp(-E_a/RT) \qquad (6)$$

where $A$ is a pre-exponential factor, $E_a$ is the activation energy, $R$ is the gas constant, and $b_T$ is the vertical shift factor. Remarkably, the activation energy for the Maxwell relaxation times is typical of linear covalent polymers[62] as shown in Supplementary Table 1. Stern and Tobolsky have shown that the activation energy of the stress relaxation in polysulfide rubbers is approximately the same despite different chemical structures indicating that the same type of bond is responsible for the decay[35,36]. Their tabulated energies range between 85 and -108 kJ/mol. Tobolsky et al. have shown that the bond dissociation energy of S-S bonds in methyl tetrasulfide (a small molecule model system) is approximately 150 kJ/mol which is similar to previous reports of typical activation energies of -146 kJ/mol[63]. While it has been shown that the flow activation energy of a polymer can differ from small molecule studies[10,64], there is an order of

magnitude difference between our energies. This could potentially be due to the higher rank of sulfur in our materials or the amount of oligomeric cyclized sulfur in the polymer.

The vertical shift factors ($b_T$) are found to vary significantly in a very small temperature range. Since the vertical shift factors typically do not vary this much in carbon-based polymers (Supplementary Fig. 27 for polystyrene), it was hypothesized that these too, in addition to the horizontal shift factors, are due to the S-S bond exchange reactions. Plots of the vertical shift factors versus temperature (Supplementary Figs. 37–42) were also modeled using an Arrhenius equation to determine the activation energy and pre-exponential factor in Eq. 6 with results shown in Supplementary Table 1. It is clear this activation energy is for some systems a factor of two or more higher compared to that for the relaxation times and although hypothesized to be affected by S-S bond exchange, its contribution to each factor must be different. The vertical shift factors for linear polymers are typically near one, so, this demonstrates another difference our systems have to traditional polymer melt rheology. Indeed, Cates and co-workers[21,23,38] do not predict such an effect for dynamic covalent bonds in a linear polymer system. At this point we can only make an observation of this effect and cannot hypothesize why this behavior occurs.

## Methods

### Materials
Elemental sulfur was acquired from Sigma-Aldrich, 1,3-diisopropenylbenzene was acquired from TCI America, 2,5-Norbornadiene was acquired from ThermoScientific, and polystyrene (Styron 685D) was acquired from AmSty.

### General inverse vulcanization synthetic procedure
Elemental sulfur (in the desired wt%) was added to a 20 mL glass vial equipped with a magnetic stir bar and was heated to 165 °C in an oil bath until a clear yellow molten phase was formed. Norbornadiene (NBD2) was then added to the molten phase (in the desired wt%). 1,3-diisopropenylbenzene (DIB) (in the desired wt%) was then injected into the molten phase via a syringe. The resulting mixture was stirred at 165 °C for 10-15 minutes (depending on exact composition) until stirring stopped due to the increased viscosity of the reaction mixture. After cooling to room temperature, the polysulfide was extracted from the vial to yield the polysulfide. Yields and exact procedures for each composition are in the Supplementary Information Methods.

### Thermogravimetric Analysis (TGA)
A TA Instruments Discovery TGA 5500 (New Castle, Delaware) was used for thermal analysis at a constant ramp rate of 10 °C/min from 30 to 800 °C. Platinum sample pans were used with approximately 10 mg of material.

### Differential Scanning Calorimetry (DSC)
A TA Instruments Discovery DSC 2500 (New Castle, Delaware) equipped with a RCS 40 cooler was used for thermal analysis. The samples of approximately 5 mg were sealed hermetically in aluminum Tzero pans. A heat-cool-heat cycle was run on each material at a ramp rate of 10 °C/min from −10 °C to 150 °C and the second heat was used for analysis of $T_g$.

### Rheological experiments
Small amplitude oscillatory rheology was performed using a TA Instruments (New Castle, Delaware) ARES-G2 rheometer. Tests were conducted using 8 mm stainless steel parallel plates with a forced convection oven accessory for temperature control above ambient. Cool nitrogen gas from liquid nitrogen was blown through the convection oven to access subambient temperatures. When necessary, 320 grit 3M sandpaper was attached to the 8 mm parallel plates to acquire measurements free of slip at higher frequencies at low temperatures. Polymer, in powdered form following synthesis, was added to the bottom plate utilizing a polymer melt ring. After allowing the material to soften and completely relax at an elevated temperature, the ring was removed, the material was trimmed, and the test gap was set. At each temperature, the linear viscoelastic region (LVR) of the polymer was determined by performing strain sweeps from 0.01 to 100% at a constant frequency of 100 rad/s. Frequency sweeps were then performed from 100 to 0.1 rad/s within the LVR at varying temperatures from low to high. Data collected under the minimum torque of the instrument are excluded.

### Solid-state Nuclear Magnetic Resonance (NMR) Spectroscopy
The synthesized material was ground to a fine powder by pressing it lightly with a pestle in a mortar. Approximately 30 mg of fine powder was loaded into a 4 mm zirconia rotor with Kel-F cap. $^{13}$C observed at 125 MHz, in 12 T magnetic field using a Varian spectrometer. Irradiation powers were 4.0 ($^{13}$C) and 3.0 ($^{1}$H) microseconds for 90° pulses. 800-3200 scans were accumulated with spinal $^{1}$H decoupling during acquisition (20 ms) and 1 s recycle delay. Observations were made at 7 and/or 12 kHz spin rate. Magic angle was set using the outer spinning sidebands of NaNO$_3$. Chemical shifts referenced externally to TMS (0.0 ppm) using the amide carbon signal of glycine (176.3 ppm). RF pulse powers and cross-polarization (CP) were verified using a glycine standard. $^{13}$C chemical shifts were predicted using ACD Labs suite 2018 with a maximum expected error of +/-3 ppm.

## Data availability
All data necessary to support the conclusions of this paper are available within the paper and in the Supplementary Information, including detailed material synthesis and characterization. All other data that support the findings of this study are available from the corresponding authors upon request.

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

## Acknowledgements

J.P. and M.M. disclose support for the research of this work from the National Science Foundation (NSF) [PFI-RP 1940942] and the Air Force Research Laboratory through DMREF-2118578, the Air Force Research Laboratories [FA8650-16-D-5404], and ENI S.p.A.

## Author contributions

D.B. performed rheological characterization, thermal analysis, data analysis, and wrote the manuscript. T.L. performed thermal analysis and material synthesis. K.K. and J.M. performed material synthesis and 1H & 13C NMR analysis. W.P. performed ssNMR characterization and analysis. J.P. and M.M. contributed to data analysis and wrote the manuscript.

## Competing interests

The authors declare no competing interests.
