## [Peer Review File · Nature Communications]

Unraveling the Rheology of Inverse Vulcanized PolymersReviewers' Comments:

Reviewer #1:

Remarks to the Author:

The authors synthesized unique polymeric materials containing organopolysulfides with a dynamic polymer network with dissociative bonds at elevated temperatures. This study demonstrated that through control of chemical composition, the polysulfides behave as thermoplastic, living polymers, or dissociative CAN. A rheological study was conducted on varying sulfur composition, organic co-monomer ratio, and polymer architecture. SAOS rheology illustrates classical thermoplastics or dissociative CANS by altering chemistry, which is an important milestone in CAN community.

Some of the noteworthy findings in this manuscript include:

- Rheological data indicates that certain compositions are Maxwellian and "living" polymers, important experimental realizations of these rheological models.
- Rheological master curves - time-temperature superposition was constructed using relaxation times as non-arbitrary horizontal shift factors.

Overall, this comprehensive study containing novelty on polymers and rheology of these complex/dynamic materials classes studied in this manuscript by combining experiments and analytical expression adds significant advances to the CAN field.

The manuscript is well written, and with most of the supporting data are provided; hence the manuscript is worthy of the broad readership of Nature Communications. A few minor comments are noted below.

1. Merging two figures, Figures 9 and 10, is recommended to illustrate better the correlation of M_x on T_g and T_{deg} and the chemical compositions.
2. It is important to show T_g from the Storage modulus and how that influence is based on chemical compositions, and M_x .
3. Need to show (Supporting info) DMA plot showing flat rubbery plateau and where the value is picked up to calculate M_x .
4. What is the cross-linking density of these systems? In the manuscript, it is stated that altering cross-linking density, however, no detailed discussion is provided. Recommend doing swelling test.

Reviewer #2:

Remarks to the Author:

This is a very well written paper exploring the thermal and rheological properties of polymers with varying amounts of S-S sulfur bonds combined with different monomers and leading to covalent adaptable networks and "living" polymers formulation. The paper shows that these polymers behave as Maxwellian polymers at high temperatures. Globally, this review is very positive and I recommend publication.

This work is significant and well justified.

The introduction is very didactic and clear.

The methodology is sound and all tests and results are well presented.

This reviewer particularly appreciated the discussion of the need for vertical shift factors, which is always somewhat controversial.

All methods are well detailed and material synthesis is clear.

One note from the reviewer is that there is extensive supplemental data cited in the paper, which renders the reading and following of arguments more difficult. Please consider including some results and tables in the paper directly if possible.

Some small comments:

L53. It seems a little strange to include values for slopes and moduli when Fig. 1 does not have numerical axes.

L.87 dominates and exhibits

L105. Pyun et al is missing a reference number

L.115 Mackay and Pyun et al. is missing a reference too.

L.137 Pyun et al. missing reference.

L.237 possesses

L.243 due to

L. 272 organopolysulfides

L.422 has -> have

Reviewer #3:

Remarks to the Author:

The authors present results on inverse vulcanization to synthesize high sulfur content (≥ 50 wt%) polymers that follow the Maxwell model of viscoelasticity (just one relaxation time), with further improvements introduced via a dimeric crosslinker.

The results are interesting but there are several problems based on which I don't think that the paper is publishable in its current form.

- it's not clear how the standard viscoelastic limit of a solid is obtained by varying the physical parameters. I.e. by increasing the crosslinking, the limit of Zener solid should be reached, with an elastic plateau showing up at low frequency in the WLF-treated data. See e.g. J. Phys.: Condens. Matter 32 395402 (2020) for the conceptual framework of this crossover, and Polymer 54 (2013) 7048e7057 for the emergence of Zener-type behaviour (Fig. 7).

I believe it would be important to show how by chemical modification a Maxwellian-type material can be turned into a standard viscoelastic solid. This crossover should also be visible upon crossing the T_g towards low temperatures, which is marked by a huge increase of the low-frequency shear modulus (cfr. Kolloid-Zeitschrift volume 134, pages149–189 (1953) and, for the molecular-level theory, Phys. Rev. Lett. 110, 178002 (2013)). This type of achievement would qualify for the level sought in Nature Communications.

- the authors use the affine rubber elasticity for G' but this is true only for the high frequency plateau. In fact, the Maxwellian interpolation between viscous and elastic limits can be microscopically shown, and understood, to be a crossover from nonaffine (at low frequency) to affine (at high frequency) microscopic dynamics of the monomers, as explained in Chapter 3 of "Theory of disordered solids", Springer 2023. It would be important to discuss this type of insights.

Reviewer #1 (Remarks to the Author):

The authors synthesized unique polymeric materials containing organopolysulfides with a dynamic polymer network with dissociative bonds at elevated temperatures. This study demonstrated that through control of chemical composition, the polysulfides behave as thermoplastic, living polymers, or dissociative CAN. A rheological study was conducted on varying sulfur composition, organic co-monomer ratio, and polymer architecture. SAOS rheology illustrates classical thermoplastics or dissociative CANs by altering chemistry, which is an important milestone in CAN community.

Some of the noteworthy findings in this manuscript include:

- Rheological data indicates that certain compositions are Maxwellian and “living” polymers, important experimental realizations of these rheological models.
- Rheological master curves - time-temperature superposition was constructed using relaxation times as non-arbitrary horizontal shift factors.

Overall, this comprehensive study containing novelty on polymers and rheology of these complex/dynamic materials classes studied in this manuscript by combining experiments and analytical expression adds significant advances to the CAN field.

Authors' response: We thank the referee for careful review of our manuscript on using material composition to control the rheological behavior of organopolysulfides and providing valuable feedback.

The manuscript is well written, and with most of the supporting data are provided; hence the manuscript is worthy of the broad readership of Nature Communications. A few minor comments are noted below.

1. Merging two figures, Figures 9 and 10, is recommended to illustrate better the correlation of M_x on T_g and T_{deg} and the chemical compositions.

Authors' response: We thank the referee for this suggestion. We agree that the succinctness of the results can be improved by moving these figures closer to each other. The figures have been combined into a single figure (now Fig. 9) to more clearly correlate crosslinking density with composition and thermal properties.

2. It is important to show T_g from the Storage modulus and how that influence is based on chemical compositions, and M_x .

Authors' response: We concur with the referee's comment that DMA would be useful to correlate glass transition (T_g) trends with chemical composition and crosslinking density. In this manuscript, we have provided these exact structure-property correlations from a combination of DSC and small amplitude oscillating shear (SAOS) rheology, where the SAOS rheology provides detailed information on the plateau moduli and crosslinking density. In conjunction with the T_g values determined via DSC, the rheological results provide the same type of correlation. We acknowledge that there are differences in DSC vs. DMA measured T_g 's in inverse vulcanized polymers (Mackay & Pyun et al., ACS Macro Letters 2015, 4, 862; ACS Macro Letters 2016, 5, 1152). However, we clearly show in this manuscript that control of co/terpolymer composition enables higher T_g values and crosslinking density in the organopolysulfides, ultimately reaching T_g 's as high as 112 °C, a marked improvement over that near room temperature in early generations of inverse vulcanized polysulfides. Due to the large number of co/terpolymer compositions, full DMA experimentation of all these materials would in fact warrant a separate study. We are in the process of preparing a more

detailed study on the effect of experimental technique to find the T_g in a follow up study, and hope the referee agrees with our current narrative with SAOS rheology and DSC.

3. Need to show (Supporting info) DMA plot showing flat rubbery plateau and where the value is picked up to calculate M_x .

Authors' response: We thank the referee for pointing out the ambiguity in the manuscript on where our values for rubbery plateau modulus are derived from in the SAOS rheology master curves. We have clarified in the manuscript that the plateau modulus is marked on the storage modulus (G') plot coincident with corresponding loss modulus (G'') local minimum in this frequency regime, which is the standard interpretation in the polymer rheology literature. For the most highly crosslinked materials, a minimum in loss modulus is not observed over the range of temperatures tested and therefore the highest value for G' is used as the plateau modulus since the rubbery region was not attainable prior to material degradation or the onset of slip. The master curve plots in the SI (Supplementary Figs. S4-S15) have been annotated to show exactly where this value is found by a black star marker.

4. What is the cross-linking density of these systems? In the manuscript, it is stated that altering cross-linking density, however, no detailed discussion is provided. Recommend doing swelling test.

Authors' response: The discussion of the crosslinking density is now centered around Fig. 9b where the molecular weight between topological constraints is shown as a function of NBD2 crosslinker mass percentage. As the molecular weight between crosslinks decreases, the crosslink density increases. As mentioned in the main text, rubber network theory provides values for this molecular weight that are nonsensical (too small on the order of a few atoms) due to the lack of S-S bond dynamicity being included in the theory. Our findings demonstrate that this class of dynamic material necessitates the development of a new framework to describe the crosslinking density (related to the comments from reviewer #3 that suggest a more detailed theoretical/microscopic interpretation of the results primed for future work). While a swelling test would be a good way to demonstrate that we indeed have a crosslinked network topology, there are no known solvents that can readily solubilize sulfur copolymers containing NBD2. In general, the "best good solvent" for inverse vulcanized polysulfides is carbon disulfide (CS_2), however, CS_2 is not chemically stable and does not readily dissolve NBD2 containing polysulfides. The ^{13}C CP-MAS NMR spectroscopy of NBD2 containing sulfur copolymers is included (Supplementary Fig. S1) to confirm that NBD2-containing polymers are crosslinked.

Reviewer #2 (Remarks to the Author):

This is a very well written paper exploring the thermal and rheological properties of polymers with varying amounts of S-S sulfur bonds combined with different monomers and leading to covalent adaptable networks and "living" polymers formulation. The paper shows that these polymers behave as Maxwellian polymers at high temperatures. Globally, this review is very positive and I recommend publication.

This work is significant and well justified.

The introduction is very didactic and clear.

The methodology is sound and all tests and results are well presented.

This reviewer particularly appreciated the discussion of the need for vertical shift factors, which is always somewhat controversial.

All methods are well detailed and material synthesis is clear.

Authors' Response: We thank you the referee for a positive and constructive review of our manuscript.

One note from the reviewer is that there is extensive supplemental data cited in the paper, which renders the reading and following of arguments more difficult. Please consider including some results and tables in the paper directly if possible.

Authors' Response: We wholeheartedly agree with the referee's suggestion that the Supporting Information (SI) is quite dense, and the manuscript narrative would be improved if more of the SI content could be re-inserted into the main text. However, the journal has a strict word count and figure limit, which has led to current formatting and SI content. We hope the referee will understand these writing constraints and accept the revised manuscript and SI, which we have endeavored to be as succinct as possible to guide the readers through these results.

Some small comments:

L53. It seems a little strange to include values for slopes and moduli when Fig. 1 does not have numerical axes.

L.87 dominates and exhibits

L105. Pyun et al is missing a reference number

L.115 Mackay and Pyun et al. is missing a reference too.

L.137 Pyun et al. missing reference.

L.237 possesses

L.243 due to

L. 272 organopolysulfides

L.422 has -> have

Authors' response: Per the referee's careful annotations above, we have added numerical values to the plots, fixed the typos, added references, and corrected typos in the references section.

Reviewer #3 (Remarks to the Author):

The authors present results on inverse vulcanization to synthesize high sulfur content (≥ 50 wt%) polymers that follow the Maxwell model of viscoelasticity (just one relaxation time), with further improvements introduced via a dimeric crosslinker.

The results are interesting but there are several problems based on which I don't think that the paper is publishable in its current form.

Authors' response: We thank the referee for careful and insightful review of our manuscript. The application of a microscopic perspective of the rheological results in this manuscript content was appreciated and incorporated where conceptual junctions to molecular level models of dynamic covalent polymers was possible. We note that the application of affine & non-affine rheological deformations has not been formally modeled for this fairly new class of polymers, which would require a separate, more detailed study to

properly describe.

- it's not clear how the standard viscoelastic limit of a solid is obtained by varying the physical parameters. I.e. by increasing the crosslinking, the limit of Zener solid should be reached, with an elastic plateau showing up at low frequency in the WLF-treated data. See e.g. J. Phys.: Condens. Matter 32 395402 (2020) for the conceptual framework of this crossover, and Polymer 54 (2013) 7048e7057 for the emergence of Zener-type behaviour (Fig. 7).

Authors's response: A low frequency plateau, sometimes referred to as an equilibrium modulus, emerges in classical thermosetting polymer networks due to the physical constraints of polymer segments between static crosslinks, preventing relaxation across long time scales. It is important to note that the materials presented in this report, have distinctive rheological relaxation processes in comparison to classical synthetic polymers due to the presence of dynamic S-S bonds between crosslinks. These dynamic bonds allow for network topological rearrangement and therefore stress relaxation at longer time scales when the bond exchange reaction is thermally activated, hence, the limit of Zener solid cannot be reached in this class of organopolysulfides. This behavior is quite different than classical thermoset polymers with static covalent bonds, where higher crosslinking density arrests viscoelastic behavior. Hence, in this report, we include a wide breadth of sulfur co/terpolymer compositions to represent the full spectrum of possible rheological behavior. In particular, we included the poly(S_{50-r}-Sty₅₀) as an example of a typical thermoplastic material that achieves terminal behavior at lower temperatures below the onset of S-S bond exchange reaction at higher temperatures.

I believe it would be important to show how by chemical modification a Maxwellian-type material can be turned into a standard viscoelastic solid. This crossover should also be visible upon crossing the T_g towards low temperatures, which is marked by a huge increase of the low-frequency shear modulus (cfr. Kolloid-Zeitschrift volume 134, pages 149–189 (1953) and, for the molecular-level theory, Phys. Rev. Lett. 110, 178002 (2013)). This type of achievement would qualify for the level sought in Nature Communications.

Authors's response: We thank the reviewer for this set of comments. In these sulfur-based polymers, compositional tuning from viscous fluids to networked polymers is described and achieved by navigation of the small subset of organic comonomers amenable to the inverse vulcanization process which is limited to those comonomers that are miscible with liquid sulfur. In our current study, we demonstrate that the incorporation of NBD2 monomer increases the crosslinking density leading to shorter sulfur rank (e.g., # of S-S bonds between organic monomer units). Maxwellian behavior is obtained despite the difference in comonomer structure leading to either a linear or crosslinked architecture. The referee's suggestion to demonstrate organopolysulfide tunability from Maxwellian type relaxation to a Zener-type standard viscoelastic behavior is well-taken. To explicitly show the chemical compositional control (as opposed to architectural control via NBD2), we have synthesized a new series of linear terpolymers from sulfur, DIB and Sty. We show that in the terpolymer poly(S_{50-r}-DIB_{40-r}-Sty₁₀) with a relatively high T_g of 17 °C, Maxwellian behavior is obtained (see new Supplementary Figs. S12 and S24). While maintaining a linear architecture, the T_g is decreased by adding more styrene comonomer (see new Supplementary Fig. S3c for DSC results), resulting in the departure from Maxwellian behavior and the recovery of a standard viscoelastic thermoplastic material (see new Supplementary Figs. S13 and S14). Hence, by altering the termonomer feed ratio in poly(S-r-DIB-r-Sty) terpolymers, control over the T_g is achieved in the linear polymers allowing control over the rheological behavior where Maxwellian behavior reverts to classical thermoplastic behavior.

- the authors use the affine rubber elasticity for G' but this is true only for the high frequency plateau. In fact, the Maxwellian interpolation between viscous and elastic limits can be microscopically shown, and understood, to be a crossover from nonaffine (at low frequency) to affine (at high frequency) microscopic dynamics of the monomers, as explained in Chapter 3 of "Theory of disordered solids", Springer 2023. It would be important to discuss this type of insights.

Authors's response: We thank the reviewer for this useful insight. It is true that a transition from affine deformations (high frequency) to nonaffine deformations (low frequency) occurs in Maxwell materials. The

reptation of polymer chains are strictly nonaffine as are S-S bond exchange events leading to topological rearrangements. This transition is therefore observed in the Maxwellian materials presented here. In this initial report on the macroscopic observation of Maxwellian behavior in organopolysulfides, the focus is placed on the phenomenological results that are controlled by polymer composition. It would appear that this type of microscopic framework has yet to be applied to dynamic covalent polymers, particularly those where thermoreversible S-S bonds dominate the rheological stress relaxation process. This in depth analysis of the microscopic dynamics of the polymeric materials is largely beyond the scope of this manuscript, however, this type of interpretation of results would be a worthwhile undertaking in a future manuscript and would represent a significant extension of the work presented in the current manuscript. However, we have included a reference on the appeal for applying affine & non-affine deformation theory to covalent adaptable network polymers and called this out in the manuscript as a new inserted sentence (Reference #61 located immediately preceding Fig. 9).

Reviewers' Comments:

Reviewer #1:

Remarks to the Author:

The authors have done an excellent job of answering questions to all referee comments. The manuscript can be accepted in its current form.

Reviewer #2:

Remarks to the Author:

The paper seems to contain some formatting issues. Nonetheless, the response to the reviewers comments were satisfying. I recommend to publish.

Reviewer #3:

Remarks to the Author:

The authors did a very nice job in addressing all the queries of the referees, including additional data and analysis. I believe that after such substantial revision and effort, the paper can be published in Nature Communications.